# Efficient Temporal Denoising for Improved Depth Map Applications

**Pengzhi Li** [1]   **Zhiheng Li** [1]

[1] Tsinghua University
{lpz21@mails,zhhli@mail}.tsinghua.edu.cn

## Abstract

Depth estimation involves acquiring three-dimensional information from images, which has numerous applications in downstream tasks. Although several effective monocular depth estimation algorithms have been developed, directly applying frame-by-frame depth estimation can result in flickering, which hinders many video-related applications. Previous video-based approaches have primarily been post-processing methods that utilize spatial information about camera poses to reduce flicker, but they come with a considerable computational cost. In this paper, we introduce the concept of depth map noise to better understand flicker in depth maps and propose a depth noise smoothing network to eliminate visual flicker in depth maps. Our approach can be applied to different depth estimation models and run in real-time for screen-based applications, such as video bokeh.

## 1    Introduction

In recent years, there has been a rapid advancement in single-frame image depth estimation algorithms Li & Snavely (2018); Li et al. (2019); Ranftl et al. (2020); Walia et al. (2022); Watson et al. (2021). Obtaining continuous depth from video is a challenging task, and many hybrid methods Luo et al. (2020); Kopf et al. (2021); Zhang et al. (2021); Teed & Deng (2020) have been proposed to address this issue by leveraging both geometric cues and learning-based priors. While this approach ensures geometric consistency, it requires high accuracy in camera poses and has a high computational overhead, making it challenging to apply in real-world applications.

Therefore, we propose a novel approach that can produce stable depth maps while minimizing computational overhead. Inspired by image denoising methods Pang et al. (2021); Moran et al. (2020), we use a post-processing approach to interpret depth map noise as inconsistencies in depth values across multiple frames. We develop a depth map denoising network that eliminates these inconsistencies by averaging the differences between multiple frames of depth maps.

The network consists of two substreams, one for predicting a scale map and the other for predicting a shift map. It is designed to learn the spatio-temporal patterns of depth map noise and generate corresponding corrections. Finally, the depth differences are averaged by multiplying and adding them with the original depth map to obtain a final, noise-free depth map. Our approach can be applied to multiple single-frame depth estimation models.

Overall, our proposed method provides a simple yet effective solution for obtaining continuous depth from video without sacrificing computational efficiency. This approach has the potential to improve downstream applications that rely on video depth maps, such as augmented reality (AR) Liu et al. (2021)and bokeh rendering Peng et al. (2022).

## 2    Method

We design a recurrent structure based on the property that depth values can be decomposed into two components: scale and shift. To achieve this, we use two consecutive frames of the depth map as input to our network and split the output into two sub-streams: a scale map and a shift map. These matrices are then multiplied and added element-wise with the latter frame of the input. This process

| Method | AbsRel $\downarrow$ | SqRel $\downarrow$ | RMSE $\downarrow$ | LogRMSE $\downarrow$ | $\delta_1 > 1.25\uparrow$ | $\delta_2 > 1.25^2\uparrow$ | $\delta_3 > 1.25^3\uparrow$ | $T_{\text{warp}} \downarrow$ |
|---|---|---|---|---|---|---|---|---|
| LeReS Yin et al. (2021) | 0.451 | 2.407 | 5.413 | 0.641 | 0.367 | 0.605 | 0.736 | 0.144 |
| DPT Ranftl et al. (2021) | 0.492 | 3.973 | 5.144 | 0.524 | 0.452 | 0.659 | 0.781 | 0.131 |
| MiDaS Ranftl et al. (2020) | 0.343 | 2.302 | 4.872 | 0.467 | 0.485 | 0.718 | 0.826 | 0.135 |
| Ours-LeReS | 0.450 | 2.402 | 5.421 | 0.612 | 0.372 | 0.616 | 0.740 | 0.069 |
| Ours-DPT | 0.401 | 2.301 | 4.923 | 0.501 | 0.464 | 0.672 | 0.801 | 0.057 |
| Ours-MiDaS | **0.334** | **2.071** | **4.870** | **0.464** | **0.496** | **0.721** | **0.828** | **0.052** |

Table 1: Quantitative comparison of depth on Sintel dataset Butler et al. (2012). **Bold** figures indicate the best and underlined figures indicate the second best.

can be defined by:

$$D'_t = Scale * D_t + Shift, (Scale, Shift) = Net(\{D_{t-1}, D_t\}) \qquad (1)$$

We use a depth domain loss Ranftl et al. (2020) as a constraint in the depth domain. Additionally, to further reduce the depth variability between adjacent frames, we employ a temporal loss, as illustrated below:

$$\mathcal{L}_{TC} = \sum_{t=1}^{T} \sum_{i=1}^{N} M_{t \Rightarrow t-1}^{(i)} \left\| D_t - \hat{D}_{t-1} \right\|_1, \qquad (2)$$

where $M$ is $\exp(-\alpha \| R_t - \hat{R}_t \|_2^2)$, we set $\alpha = 50$. $\hat{D}_{t-1}$ represents the warped frame $D_{t-1}$. $\hat{R}_t$ is the warped RGB frame by the backward optical flow $F$. During training, we compute a dense optical flow $F$ using FlowNet2 Ilg et al. (2017).

Our network comprises two convolutional layers, five residual blocks, and a transposed convolutional layer that is split into two sub-streams, each of which outputs a different map: the scale map and the shift map. The sub-stream for predicting the scale map is augmented with a skip connection from the encoder to the decoder. The training dataset consists of disparity maps and RGB images obtained from 3D movie processing, which are transformed into the depth domain while excluding invalid depth regions. We use a batch size of 4 and input a sequence containing 7 frames. Video frames are cropped to 384 × 384. The initial learning rate is 1e-4 and reduced by half every 10,000 iterations.

## 3 EXPERIMENTS

We assess the feasibility of our approach on the Sintel dataset Butler et al. (2012) using standard metrics for evaluating depth estimation, the table 1 presents the corresponding quantitative results. Our method improves depth accuracy of three depth estimation models. Additionally, we select a few real-world scene sequences for testing, as depicted in Appdenix. Our approach effectively eliminates depth noise between frames and reduces visual flicker.

In order to assess the temporal stability of the depth maps, we employ the calculation of flow warping error between two frames. The warping error is calculated as:

$$T_{\text{warp}} = \frac{1}{T-1} \sum_{t=1}^{T-1} \left( \frac{1}{\sum_{i=1}^{N} M_t^{(i)}} \sum_{i=1}^{N} M_t^{(i)} \| D_t^{(i)} - \hat{D}_{t+1}^{(i)} \|_2^2 \right), \qquad (3)$$

where $\hat{D}_{t+1}$ represents the warped frame $D_{t+1}$, and $M_t$ is a binary mask that indicates non-occluded regions. The warping error is evaluated by computing the average warping error across the entire sequence.

## 4 CONCLUSION

Our approach is motivated by practical applications, as we prioritize the application of screen editing over scene reconstruction and localization. The proposed method is based on the concept of depth maps denoising and represents the first post-processing method capable of effectively alleviating depth flickering in videos in real-time. This method facilitates video editing tasks that require the use of depth maps straightforwardly and efficiently, which can lead to more accurate and visually appealing results.

URM STATEMENT

The authors acknowledge that at least one key author of this work meets the URM criteria of ICLR 2023 Tiny Papers Track.

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

## A  APPENDIX

We present optimization results for some real-world depth maps in  fig. 1.

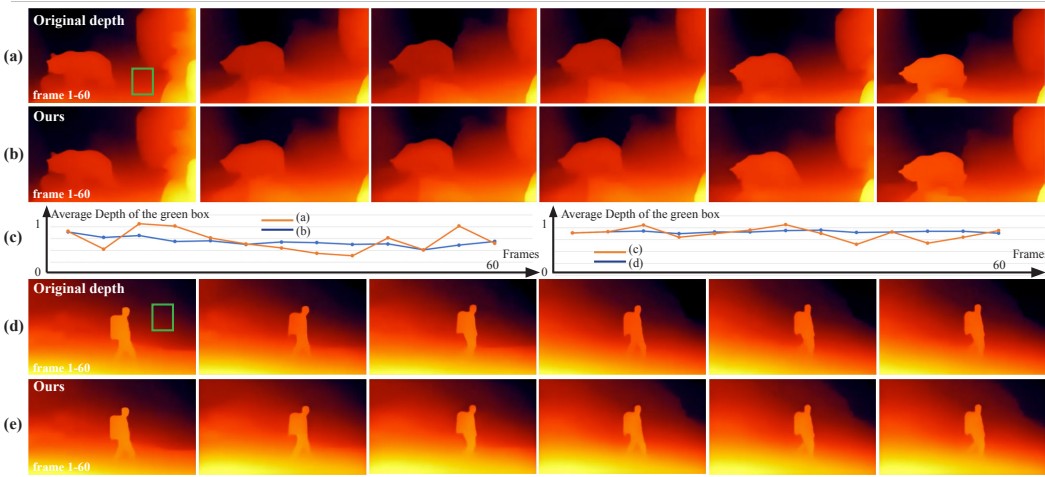

Figure 1:  Comparisons of real results. (a) and (d) depict the initial depth maps, while (b) and (e) depict the optimized depth maps. The line plot in (c) represents the average depth within the green box in the video, where smoother lines indicate greater consistency between the depth maps.

