# OpenReview forum: "Efficient Temporal Denoising for Improved Depth Map Applications"
_ICLR.cc/2023/TinyPapers — Submitted to Tiny Papers @ ICLR 2023_

### Official Review · Reviewer_BVRd · 2023-03-27

**Confidence:** 4

**Summary Of Contributions:**

The paper presents a simple yet effective approach for temporal denoising of depth maps estimated from monocular images. The method consists of a CNN which is trained with a depth prediction and a temporal loss, and is tested on a public dataset.

**Rating:**

High Potential (HP): a submission which meets the reviewing criteria and has potential to make an impact on the field

**Strengths And Weaknesses:**

The proposed method leverages an existing approach modelling depth as disparity up to scale and shift and introduces an additional temporal smoothing loss. The method is tested on a public dataset and the quantitative and qualitative results show that the proposed method effectively improves the depth map quality.
The paper is clear and easy to follow, and the English level is satisfactory.

In my opinion, the main weakness of the paper is the lack of description of how this method relates to the image denoising and the depth map noise concept in Section 2. The relationship is mentioned a few times, but never explained clearly.
Another significant weakness is the lack of comparison with other depth denoising/smoothing methods (even baselines such as temporal gaussian smoothing) and the lack of an ablation study on the proposed temporal loss and some of the hyper-parameters (e.g. length of the sequence).

Overall, the approach seems efficient and could have the potential for improving real-time applications that require temporally consistent depth map estimation, such as video editing.

**Suggested Changes:**

The revised paper could:
- clarify the relationship between this method and the image denoising/depth map noise;
- clarify which loss is used as depth domain loss among those presented in Ranftl et al. (2020);
- include computational cost information and potentially a comparison with geometrical methods;
- fix the citation format, using \citet for in-text citations (e.g. "Peng et al. (2022) showed that") and \citep for other citations (e.g. "denoising methods (Pang et al., 2021)");
- in Tab. 1, reorder the rows. I believe that ordering the lines as method 1, method 1 + ours, method 2, method 2 + ours, etc. would be more effective in showing when/how much the proposed approach improves the estimated depth (columns with percentage improvements could be effective too);
- in Fig. 1, add a row containing a zoomed-in salient area of the frames to highlight the differences between original and ours (e.g. on the legs of the walking person);
- in Fig. 1 (c), review the legend of the graph or clarify the meaning of the two lines in the caption (for instance, what (c) stands for?);
- include a link to a video showing a comparison between the original and the denoised depth map.

---

### Author Response · Authors · 2023-05-30
**Opt-in for archival**

We wish to opt-in for archival.

---

### Meta-Review · Area_Chair_JJFy · 2023-04-08

**Recommendation:** Invite to present
**Confidence:** 4

**Metareview:**

Based on the review, the paper proposes a simple but effective approach for temporal denoising of depth maps estimated from monocular images. The method uses a CNN trained with a depth prediction and a temporal loss. The strengths of the paper include the clear presentation of the method and the positive results obtained in the experiments on a public dataset. However, the paper has some weaknesses, such as the lack of explanation of the relationship between the method and image denoising/depth map noise, the lack of comparison with other methods, and the absence of an ablation study on some of the hyperparameters. The paper could be improved by addressing the weaknesses mentioned in the review, such as clarifying the relationship with image denoising, including computational cost information, and adding a comparison with other methods.

Pros:
- Simple and effective approach
- Clear presentation of the method
- Positive results on a public dataset

Cons:
- Lack of explanation of the relationship with image denoising/depth map noise
- Lack of comparison with other methods
- Absence of an ablation study on some hyperparameters


**Summary:**

The paper presents an approach for temporal denoising of depth maps estimated from monocular images. The method consists of a CNN which is trained with a depth prediction and a temporal loss, and is tested on a public dataset.

**Reason For Not Giving A Higher Recommendation:**

 It can be further improved when incorporating the suggested changes by the reviewer.


**Reason For Not Giving A Lower Recommendation:**

Overall, this paper is a good submission. Please refer to the strength sections.

---

### Decision · Program_Chairs · 2023-04-08

Invite to present